# Nanobionics: A Sustainable Agricultural Approach towards Understanding Plant Response to Heavy Metals, Drought, and Salt Stress

**DOI:** 10.3390/nano13060974

**Published:** 2023-03-08

**Authors:** Mohammad Faizan, Fadime Karabulut, Pravej Alam, Mohammad Yusuf, Sadia Haque Tonny, Muhammad Faheem Adil, Shafaque Sehar, S. Maqbool Ahmed, Shamsul Hayat

**Affiliations:** 1Botany Section, School of Sciences, Maulana Azad National Urdu University, Hyderabad 500032, India; 2Department of Biology, Faculty of Science, Firat University, Elazig 23119, Turkey; 3Department of Biology, College of Science and Humanities, Prince Sattam bin Abdulaziz University, Alkharj 11942, Saudi Arabia; 4Department of Biology, College of Science, United Arab Emirates University, Al Ain 15551, United Arab Emirates; 5Faculty of Agriculture, Bangladesh Agricultural University, Mymensingh 2202, Bangladesh; 6Zhejiang Key Laboratory of Crop Germplasm Resource, Department of Agronomy, College of Agriculture and Biotechnology, Zhejiang University, Hangzhou 310058, China; 7Department of Botany, Faculty of Life Science, Aligarh Muslim University, Aligarh 202002, India

**Keywords:** heavy metal, nutritional demands, physiological traits, nanotechnology

## Abstract

In the current scenario, the rising concentration of heavy metals (HMs) due to anthropogenic activities is a severe problem. Plants are very much affected by HM pollution as well as other abiotic stress such as salinity and drought. It is very important to fulfil the nutritional demands of an ever-growing population in these adverse environmental conditions and/or stresses. Remediation of HM in contaminated soil is executed through physical and chemical processes which are costly, time-consuming, and non-sustainable. The application of nanobionics in crop resilience with enhanced stress tolerance may be the safe and sustainable strategy to increase crop yield. Thus, this review emphasizes the impact of nanobionics on the physiological traits and growth indices of plants. Major concerns and stress tolerance associated with the use of nanobionics are also deliberated concisely. The nanobionic approach to plant physiological traits and stress tolerance would lead to an epoch of plant research at the frontier of nanotechnology and plant biology.

## 1. Introduction

Sensitive plants must develop sophisticated defence mechanisms against environmental disruptions, especially abiotic stress which reduces plant productivity. Due to such disruptions in plant life and the growing global population, there is a significant imbalance in the production and consumption of food [1]. In order to balance crop productivity and food demand, it is now necessary to develop effective techniques for acclimating plants to stressful situations. Abiotic stress such as HMs, drought, and salt can be harmful to plants, but when they are present in small concentrations and for brief periods of time, they can activate a variety of signalling cascades and pathways in plants. As a result, plant life becomes more resilient to future stressors [2]. The limitations of these methods that activate the plant defence system in response to an environmental stress has heightened the urgency of finding a powerful and cost-effective alternative to demonstrate plant tolerance to a variety of stress conditions [3]. Exposing plants to either natural or synthetic chemical preparation agents has become a major focus of plant stress physiology and stress management researchers in recent years. In order to achieve this, they are working to develop plant varieties that are more tolerant to a variety of environmental stressors [4,5]. Currently, a method known as nanopriming uses nanoparticles (NPs) as a primer agent to improve seed germination, seedling growth, and stress tolerance, which increase plant yield and food quality [6]. The advantages of seed nanopriming are associated with a variety of biochemical, physiological, and signalling pathways in seeds, including the preservation of reactive oxygen species (ROS) homeostasis and their interference with phytohormones [7,8].

The study of objects on a nanometer (nm) scale, where properties enable new and wide uses, is known as nanoscale science and nanotechnology. Therefore, whether they are produced artificially or naturally, materials that are at least one dimension smaller than 100 nm are referred to as nanomaterials (NMs). To name a few examples, engineered NMs can be single, fused, or agglomerated particles, spherical or nearly spherical particles, tubular or irregular (non-spherical) particles, and they could have either a homogeneous or heterogeneous composition [9]. Research aimed at effectively using agrochemicals has advanced thanks to nanotechnology [10]. Scientists are primarily focused on creating delivery platforms to encourage, control, or delay the release of fertilizers, pesticides, and phytohormones in the early stages of nano-enabled agricultural research [11,12]. The platforms include engineered nanoscale hydrogels, membranes, fibres, particles, biodegradable polymers, lipids, proteins, and minerals. It is typical to use an active ingredient (cargo) that has been chemically or physically enclosed in a carrier. These platforms, in the first place, will make it possible to control the cargo’s dissolution profiles, preventing early physicochemical deterioration [13]. With the development of genetically modified organisms (GMOs) and other more advanced farming techniques such as indoor, vertical, and hydroponic farming, projected needs to improve crop production are sought to be made more efficient and specific [14]. Particle adhesion and uptake, as well as particle characteristics (size, chemistry, geometry, etc.), are taken into consideration when designing particle delivery systems for agricultural applications [15]. Consequently, it has a big impact on how particles move and gather in plants. These systems aim to meet the demand in an effort to further improve control over the release of agrochemicals into the environment. This will result in noticeably higher crop yields and significantly less contamination.

In order to permit nanopriming and seed germination—a crucial and delicate stage of plant life involving a variety of metabolic events taking place—environmental stress needs to be overcome during this stage of the life cycle. Only then will the plant move on to later growth stages [1]. In order to increase abiotic stress resistance, seeds can be pretreated with natural or synthetic substances to partially hydrate them or to mildly stress them. This process is referred to as “seed preparation” [16]. *Festuca ovina* seeds are nano-primed with silver (Ag) NPs at different concentrations (25%, 50%, and 75%) to boost germination rates and percentages to lessen the damaging effects of drought stress on seeds [17]. When compared to hydro-primed and unprimed seeds, the germination rate of *Brassica napus* L. seeds prepared with zinc oxide (ZnO) NPs significantly increased at all concentrations by reducing the effects of salinity stress [16]. Additionally, it has been found that seeds treated with coated nanoceria with 0.1 mM polyacrylic acid and subjected to salt stress underwent higher water and amylase activities and germination rates over control seeds [18]. Abou-Zeid and Ismail [19] have demonstrated that biosynthesized AgNPs (1 mg/L^−1^) enhanced the germination % and reduced the toxicity caused by salt stress in the seeds of *Triticum aestivum*. The addition of Multi-Walled Carbon Nanotubes (MWCNTs) to *Dodonaea viscosa* L. seeds caused a reduction in the mean germination time and increased seed germination when the seeds were subjected to a more intense form of drought stress [20]. It is demonstrated that the addition of silicon oxide (SiO_2_) NPs (15 mg/L^−1^) to *T. aestivum* seeds prior to germination in the conditions of drought stress leads to an increase in water uptake as well as amylase activity [21]. Additionally, salinity stress reduced the germination of *Zea mays* L. seeds when treated with 1000 mg/L^−1^ of mango peel NPs (nMPs). Furthermore, it appears to have a sizable effect on reducing high germination percentage [22]. A study found that *Daucus carota* L. cv. Mussangochon seeds treated with an AgNPs suspension (up to 2000 mg·L^−1^) exhibited a lower germination rate and there was no statistically significant difference in germination [23]. These results undoubtedly point towards the necessity of investigating the variables and mechanisms influencing seed germination via nanopriming. There is a lot of research on this topic in recent years (Table 1). Thus, the present review deals with the recent developments on applications of nanobionics to ameliorate the toxicity caused by HMs, drought, and salt stress and their possible tolerance mechanism without ignoring the potential effects of nanobionics on physiological traits, growth indices, and root architecture of crop plants.

Nanoscale objects created by humans can now be placed inside living cells, thanks to advancements in nanotechnology and nanoscience [32]. Additionally, nanotechnology has the potential to give photosynthetic organelles and organisms new and improved functional properties to capture solar energy and enhance biochemical sensing [33].

## 2. Nanobionics and Physiological Traits

NMs are functionalized into plants to improve their natural processes, such as photosynthesis. In this instance, functional NPs play a more active role in the plant than nanofertilizers do in distributing macro- or micronutrients [34]. Another goal of plant nanobionics is to make plants into gadgets such as light-emitting plants or environmental sensors. The plant is given a non-native functionality by the NMs introduced here. The basis of plant nanobionics is smart NMs that localise within tissue and even within organelles such as chloroplasts. To enter plant cells, NPs must pass through the cell wall and cell membrane [35].

By adding functionalized carbon nanotubes to plants’ chloroplasts, which are their primary photosynthetic organelles, the first application of plant nanobionics aims to increase the efficiency of their photosynthetic processes. According to improved electron transport rates, functionalized SWCNTs enter *Spinacia oleracea* leaves and boost the plant’s photosynthetic activity. The cell covering has a dielectric steady that is higher than the chloroplast. As a consequence of this, NPs with a higher charge do not induce a sufficient transmembrane potential to enter the chloroplast, but they are able to enter the cytosol. Higher potential NPs are necessary for chloroplast localization. In this study, the internalization of NPs by plant cells is discussed; however, neither the mesophyll nor the cuticle of the leaf is taken into consideration. The distribution of NPs on plant leaves can produce physical barriers and reduce the photosynthesis by degrading the chlorophyll molecules [34]. When it comes to the vast majority of plant species, the thylakoid membranes of the chloroplasts serve as the primary site of the photosynthetic mechanism. Chloroplasts are capable of absorbing the visible spectrum, which accounts to 50% of incoming solar radiation. Additionally, plants typically only use 10% of the available sunlight [36]. Consequently, researchers are working to increase the efficiency of photosynthetic processes by broadening the range of sunlight absorption [37]. Advanced and novel functional properties are produced in photocatalytic complexes based on chloroplasts using NMs with superior physical and chemical properties. Because chloroplast antenna pigments have a high rate of absorption, Single-Walled Carbon Nanotubes (SWNTs) can only take a limited amount of light in the visible and near-infrared spectrums [33]. Coating high-charged SWNTs with chitosan, biomolecules derived from the shells of shrimp and other crustaceans, allowed them to self-penetrate into chloroplasts. This has made it possible for the SWNTs to self-penetrate into chloroplasts.

CNTs or NPs can be encased in highly charged DNA or polymer molecules using the novel lipid exchange envelope penetration (LEEP) method, which also makes use of nanostructures. This has made it possible for the CNTs or NPs to travel through the hydrophobic, fatty membranes around chloroplasts. The incorporation of SWNTs into chloroplasts has the possibility to improve the light reactions that are necessary for photosynthesis due to the special optical properties that SWNTs possess. SWNTs have the ability to capture the visible and near-infrared spectra of light wavelengths, in contrast to the limited absorption rates of chloroplast pigments. Excitons are produced as a result of this solar energy being converted by SWNTs, which then transport electrons to the photosynthetic apparatus [38]. When chloroplasts are removed from plants and CNTs are added to them, the photosynthetic activity of the chloroplasts is increased by 49% when compared to the control [39]. The electron flux related to photosynthesis is increased by 30% when these nanocomposites are added to the chloroplasts of the leaves of living plants [39]. Extracted chloroplasts and leaves can be analysed with SWNT real-time NO detection to identify a variety of plant signalling molecules as well as exogenous compounds such as pesticides, herbicides, and environmental pollutants. This can be accomplished by using the leaves and chloroplasts. In order to enhance the photosynthetic environment, SWNT-based nanosensors can track the dynamics of a single free radical molecule inside chloroplasts [39]. The primary drawback of using extracted chloroplasts for solar applications is their susceptibility to degradation due to damage of photosynthetic proteins caused by light and oxygen. It shows how cerium oxide NPs (nanoceria) team up with an extremely exciting polymer, go through the external membranes of the chloroplast, and then settle in the stroma where they significantly slow down the deterioration of the photosystem by quenching the ROS [33]. These NPs can be found all over the chloroplast, enabling nanosensors to be implanted in the plant to monitor free radical species and environmental pollutants both ex vivo and in vivo [40].

Recent studies have demonstrated how to modulate the opening of the stomata in response to light using poly-3-hexylthiophene (P3HT) NPs [41]. *Arabidopsis thaliana* leaf epidermal strips are incubated in a P3HT NPs solution. Additionally, P3HT beads demonstrated that P3HT affected the oscillations of cytosolic calcium (Ca^2+^) concentration stimulated by green light at the absorbance peak (540 nm). The number and amplitude of cytosolic Ca^2+^ oscillations are significantly reduced after the polymer was excited by light. Ca^2+^ modulation served as a direct mediator in the mechanism that caused the stomata to close. As a result, we demonstrated that P3HT NPs are suitable for optically controlling Ca^2+^ concentrations and modulating stomatal functioning. Utilizing the optoelectronic characteristics of conjugated polymer NPs can be handled for the light stimulation of plant signalling as an alternative to optogenetics [38]. Zn NPs applied to sugarcane roots recently served as an example of the uptake of carriers [42]. Model plants *Oryza sativa*, *Digitaria sanguinalis*, and *A. thaliana* had their roots treated with metolachlor-loaded polyethylene glycol (PEG)-PLGA particles, solid lipid NPs (SLN), nanostructured lipid carriers, and lipid-based nanoemulsions (roughly 150 nm) via apoplastic pathways [43]. Studies in the area attested to the fact that the plant’s chloroplasts successfully internalised carbon nanotubes (high aspect ratio particles) [44].

In an effort to genetically modify the plant, DNA molecules are typically added to certain organelles to improve plant performance. It is interesting to note that until it reaches the basic pH, the pDNA–chitosan–SWCNT complex in the chloroplast preserves pDNA very well in the relatively acidic pH of leaf mesophyll. Strong ionic interactions at the pDNA–chitosan interfaces are to be blamed for this. Transmission of pDNA occurs at basic pH because of the weaker interface with chitosan [45]. This transgene distribution platform has proved to be straightforward and affordable. NPs are also being investigated for their use in bioaugmentation efforts, which involve targeted localization in plant tissues. These efforts are comparable to those involving CeO_2_. [46]. Another illustration of this concept is the construction of a peptide–DNA complex, which was performed in order to selectively target particular plant cells (plastids) for the purpose of genome engineering [47]. Here, two peptides are used: one to target chloroplast cells specifically (chloroplast targeting peptide, CTP, KH9-OEP34), and the other to encourage cell penetration (i.e., cell-penetrating peptide, CPP, BP100). The positively charged CTP peptide and the negatively charged pDNA are electrostatically complexed, and the positively charged CPP peptide is then added on the top to form the spherical nanoscale particles. CTP–pDNA–CPP particles can aggregate thanks to this design. As a result, these particles effectively translocate and transmit the DNA in the plastid as they move in the direction of the plant cell membrane. Uptake is produced by vesicle formation and intracellular trafficking. This approach to gene modification seems to work well for other plants besides model species similar to *A. thaliana*, such as *Nicotiana* and *Solanum lycopersicum* [47]. NMs can be successfully designed by carefully controlling their size and surface functionalization in order to add new functions to the living plant [48]. In other words, all designed NPs can enter in the leaf mesophyll cells because the stomatal opening is larger than 10 m. Each NP is, nevertheless, targeted to a specific tissue by their size exclusion. In the mesophile, the larger PLGA particles, which are 200 nm in size and contain luciferin, migrate into the intercellular spaces. As a consequence of this, it releases the luciferin so that it can interact with the enzyme that is released further down in the leaf tissues. In the presence of oxygen (O_2_), magnesium ions (Mg^2+^), and adenosine triphosphate, firefly luciferase is responsible for catalysing the oxidation of released luciferin, which ultimately results in the production of bioluminescence (ATP). Additionally, a concept that is applicable to agricultural sensing will be developed through this strategy [48].

## 3. Nanobionics and Growth Indices

NPs not only encourage plant growth but also protect it from the adverse effects of abiotic stress. The toxic metal (Zn, Fe, and Co) binds to the NP because of the NP’s larger surface area despite its smaller size, which reduces the metal’s availability [40]. NPs or aggregates of NPs with a diameter less than the pore diameter of the cell wall could pass through ores and can reach the plasma membrane. There is, additionally, a chance for the enlargement of pores upon interaction with NPs which, in turn, enhances the NPs’ uptake [49]. How plants respond defensively to abiotic stress has been investigated. SiO_2_ NPs, for example, increase the rate at which plants transpire, the effectiveness with which they use water, the amount of total chlorophyll, and the activity of the enzyme carbonic anhydrase in *Cucurbita pepo* as a defence against salt stress. Linolenic acid was found to be inhibited by TiO_2_ (anatase) in the electron transport chain and its photoreduction activity was also found to be changed, and it slowed down the rate of oxygen evolution in chloroplasts [50]. NP interaction and absorption causes molecular changes that have an impact on the morpho-physiology of plants. Additionally, enhanced growth and viability depend on the NPs’ capacity to pierce the tough seed coat and enable water import [51]. Furthermore, a promising tactic is used in nanotechnology to prepare seeds [52]. The growth of some significant crops, such as *Z. mays*, *Brassica juncea*, and *Glycine max*, is impacted by the application of MWCNTs. Raman spectroscopy and TEM imaging have confirmed that exposed seeds appear to have MWCNT accumulation within the endosperm [53]. NPs travel throughout the plant and engage with cellular machinery as they do so. As a result, they foster plant growth [54]. Even when mesoporous silica NPs (MSN) are present in the highest concentration (2000 mg/L), none of the plants exhibit signs of stress. This demonstrates how they can be utilised as a smarter, safer distribution system [55]. With some NPs, *Brassica napus* encourages plant growth and improves seed viability and germination [56]. Any NP’s impact can be either favourable or unfavourable depending on its size and additive concentration. In comparison to larger particles, plants accumulate more in the smallest TiO_2_ NPs [57]. Additionally, these NPs can affect the levels of miRNA to initiate growth-promoting pathways in plants [58]. Other metal oxides NPs, such as Cu and Zn, have also displayed their impact on the seed germination of *Vigna mungo* [59].

## 4. Nanobionics and Root Architecture

Root exudates could be the first obstacle to effective NP uptake in the roots of plants. It is the compound secreted by plant roots and serves an important role as a chemical attractant in the rhizosphere. It is possible to distinguish between high molecular weight substances (mostly mucilage and proteins) and low molecular weight substances (mostly organic acids, amino acids, phenolics, sugars, and various other secondary metabolites) in root secretions [60]. It has been distinguished that about 30–40% of photosynthetically fixed carbon is transferred to the rhizosphere as root exudates, including low and high molecular weight substances [61]. Plant roots can sense soil conditions and release specific organic substances in the rhizosphere. For example, oxalate, acetate, and malate in plant roots are often induced by soil nutrient-deficient conditions [61]. Plant roots possess specialized surface tissues to divide them from the environment by modifying their anatomy. Currently, very little information is available on how these NPs affect the soil microbial community. They may have an impact on soil microorganisms through direct effects, changes in the bioavailability of nutrients, indirect effects resulting from their interaction with natural organic compounds, and interactions with toxic organic compounds which would alleviate their toxicity [62,63]. Many studies have looked at how root exudates affect how well NPs are absorbed by roots. According to the study by Cervantes-Avilés et al. [64] in root exudates, nano-Cu(OH)_2_ and nano-MoO_3_ increased their aggregate size for 50 days; their mean size increased from 518 nm to 938 nm and 372 nm to 690 nm, respectively. In *G. max* root exudates, the wavelengths of nano-CeO_2_ and nano-Mn_3_O_4_, respectively, decreased from 289 nm to 129 nm and 761 nm to 143 nm. The four NPs under test are negatively charged in the exudates because the organic acids bind to the particle surfaces in the exudates from the roots [64]. Citrate in *T. aestivum* root exudates improved the dissolution of CuO NPs in the study performed by McManus et al. [65]. Since they are present in plant tissue, NPs may not be accurately evaluated based on their elemental composition. It is important to consider how quickly elements are released from NPs when they are activated by root exudates [66]. Bao et al. [67] reported that citric acid as a typical organic acid secreted by plant roots can enhance cerium uptake and accumulation in *O. Sativa* seeds when *O. Sativa* is exposed to cerium oxide NPs. It is confirmed by that study that positively charged NPs are more easily adsorbed on the surface of roots and then consumed by the plants as compared with the negatively charged NPs [68]. The other barrier to effective NP uptake by roots may be the root epidermis. The apoplastic and symplastic pathways are the main pathways for NP root uptake at the interface between NPs and the root epidermis [69]. The cell wall porosity and width of the plasmodesmata are the important aspects that influence NP root uptake efficiency for the apoplastic pathway. The main barriers for effective NP uptake along the symplastic pathway include cell wall porosity, plasma membranes, organelle membranes, as well as the diameter and the width of the plasmodesmata. The apoplastic barriers in the roots may be impacted by the NPs [70]. For instance, in *Brassica napus*, CeO_2_ NPs reduce the length of the root apoplastic barrier [71]. Early apoplastic barrier development is induced by La_2_O_3_ NPs [72]. According to this, the effectiveness of NPs being absorbed at the root may be influenced by a variety of factors. The main barriers of NP movement through the root cortical cells are mainly associated with apoplastic and symplastic pathways, and they are probably very similar to those in the root epidermis. The NPs enter the endodermis after passing through the cortex and root epidermis. Here, in addition to the endodermal cells’ cell wall porosity, the Casparian strip—a localised impregnation of the primary cell wall that surrounds the endodermal cell that is similar to an arch in the longitudinal direction—acts as a barrier for the effective uptake of NPs [66]. A quarter to a third of the transverse/anticlinal cell wall is covered by this belt, which is located in the centre of the transverse and anticlinal walls [66]. Lignin, suberin, and a few structural proteins are deposited among the substances in the Casparian strip. These substances have the ability to lower the diffusional apoplastic water flow and dissolve into the stele [73] (Figure 1). Cracks in the lateral roots’ cortex and root endodermis allow submicron particles to enter into the xylem [74]. These results imply that they might be able to enter the stele through a potential break in the Casparian strip or the effects of NPs on its characteristics. Transpiration rate may also influence how well the roots absorb the NPs once they get into the xylem [75].

There are a variety of challenges that need to be overcome, some of which depend on whether the NPs are applied as a foliar spray or as an application to the roots of the plant. This affects how far large NPs (>20 nm) are able to enter the roots, as opposed to how much of them are taken up by the stomata of leaves [76]. The cracks at the lateral root’s exit site allow sub micrometer plastics to enter the xylem, cortex, and the endodermis of the root [74]. This “crack entry” mode may be caused by potential Casparian strip discontinuity. Another potential area for NP entry is the region surrounding the root meristem. As a result of their frequent active division, stem meristem cells in this region have weaker cell walls than mature stem cells, which could make it possible for some large NPs to be transported. During the early stages of development, cuticles develop on the root caps of primary and lateral roots. These cuticles are there to protect the delicate meristems [75,77]. The nucleus, chloroplasts (also called plastids), Golgi apparatus, mitochondria, vacuoles, endoplasmic reticulum, and peroxisomes are the primary organelles that make up the plant cells. Membranes with a single layer are found around the vacuole and the peroxisome, whereas membranes with two layers can be found surrounding the nucleus, the Golgi apparatus, mitochondria, chloroplasts, and the endoplasmic reticulum. Monitoring the dietary requirements of plants is another strategy. Metal nutrients for plant growth include Mg, Zn, Mn, Cu, Ca, Fe, K, and Mb. To a large extent, allay varies or metal elements can be used to design NMs. Cucumber’s ability to withstand salinity stress is increased by Mn_3_O_4_ NPs with ROS scavenging properties. A further benefit of using NPs based on Mg, Zn, Fe, and Cu, is that they make plants more resilient to stress [78,79].

## 5. Nanobionics and Stress Tolerance

Different organelles may have different mechanisms for how NMs move across the membranes. In one study, it was discovered that ZnO NPs (46 nm, 400 mg/L) caused programmed cell death in tobacco BY2 cells by inducing oxidative stress, endoplasmic reticulum dysfunction, and mitochondrial dysfunction [80]. In stressed plants, excessive ROS accumulation and subsequent oxidative damages are common. In order to keep the balance between the production of ROS by plants and their ability to scavenge them, scientists are working on both enzymatic and non-enzymatic antioxidant systems. Utilizing NMs in order to improve plant ROS homeostasis maintenance is a viable approach that could support the development of nano-efficient agriculture. To help in the increase in plant stress tolerance and effective delivery of ROS scavenging, environment friendly NPs of plants may be a good strategy [81]. NPs enhance the photosynthetic activity of stressed plants, whereby the improvement of root growth, upregulation of aquaporins, altered intracellular water metabolism, accumulation of compatible solutes, and ionic homeostasis are the main mechanisms by which NPs alleviate stress caused by abiotic factors. NPs reduce leaf water loss caused by the accumulation of abscisic acid through the activation of antioxidant defence systems. Carbon NMs can harm plants if their properties are not properly controlled because of their small size and ease of surface conjugation [82]. Rare element-enriched agricultural products are becoming more and more popular in the market; selenium-rich agricultural products are just an example [68,83]. One strategy that is being pursued in order to increase the tolerance level of plants to abiotic stress is the development of genetically modified crop varieties with different gene paradigms. It is anticipated that these varieties will improve plant performance while they are being subjected to stressful conditions. Physiological procedures such as seed germination, seedling growth, photosynthesis, metabolism, POX, APX, CAT activities, chlorophyll content in the leaf, overall carbohydrate content, protein content, and yield increase with the application of carbon NPs. Positive gene expression changes to the possible applications in crop improvement. The capacity of roots to withstand water stress is increased by these particles as far as water uptake and hydraulic conductivity. Due to their high mobility in nature, these NPs offer crucial information for the quick delivery of nutrients to all parts of the plant [84]. For instance, applying various CNM concentrations enhances watermelon’s physiological characteristics, seedling growth, and germination rate. When compared to control treatments, such exposure boosts CAT, POD, and SOD activities, which aid in reducing a variety of abiotic stresses. Additionally, CNTs can function as an artificial antenna for the chloroplast, assisting it to absorb green light, infrared, and ultraviolet (UV) [63]. The use of polyhydroxyfulerene (PHF), a water-soluble carbon nanomaterial, can lessen the level of plant toxicity brought by HMs [43]. In a similar manner, MWCNT application in *A. thaliana* promotes plant growth by lowering paraquat toxicity [85]. With the increase in pollution in the soil, water, and air, the application of NPs as remediation towards less to no damage of surroundings is found. Bioremediation is highly time-consuming and microbe-dependent. Compound remediation, which depends on the kinetic rate of the reaction and nanobionics remediation, is largely effective, friendly in nature, and does not release hazardous by-products. The implementation of nanobionics in sustainable agriculture is thriving with the submission of nanofertilizers and nano-pesticides [86].

## 6. Conclusions and Future Prospects

Nanobionics has the enormous potential to reduce the toxicity caused by HMs, drought, and salt stress. It makes distinctive and improved functional properties of crop plants by activating defence mechanisms. It can modulate physiological traits and growth indices by adding functionalized nanomaterial to plant chloroplasts, which are their primary photosynthetic organelles. Advanced and genuine application of nanobionics aims to enhance the efficacy of the plant photosynthetic system. Consequently, a complete study needs to be conducted to spot other progressions in nanobionic employment in various environmental remediation applications.

## Figures and Tables

**Figure 1 nanomaterials-13-00974-f001:**
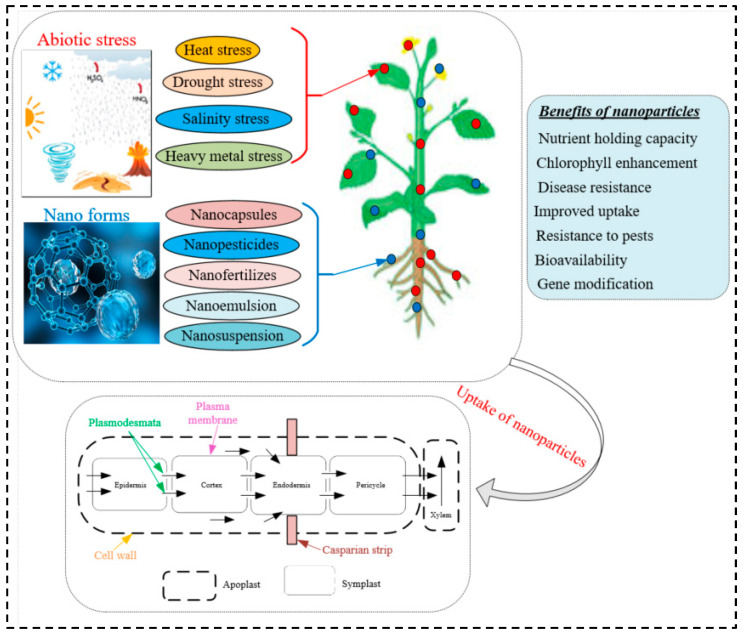
Potential interactions between NPs and the plant system are depicted schematically as a result of the interaction between HMs, drought, and salt stress.

**Table 1 nanomaterials-13-00974-t001:** The effects of NPs on the development of various plant species.

Plant Species	Applied NPs	Abiotic Stress	Concentration and Mode of Application	The Result of the Experiment	References
*Phyllostachys edulis*	SiO_2_ NPs	Cd	100 and 200 μM; seed germination	Increased seed germination rate and percentage and mean germination time	[24]
*Yuxiangyouzhan* and *Xiangyaxiangzhan*	ZnO NPs	Cd	0, 25, 50, and 100 mg/L^−1^ through seedlings	Increased plant length and fresh weight	[25]
*Zea mays* L.	CuNPs	Drought stress	3.33, 4.44, and 5.55 mg/L plant priming	An increase in the biomass, anthocyanin, chlorophyll, and carotenoid contents along with number of seeds the grain yielded	[26]
*T. aestivum*	TiO_2_	Drought	500, 1000, and 2000 mg/kg amended soil	Improved stomatal conductance, transpiration rate, antioxidative enzymes	[27]
*Abelmoschus* *esculentus* L.	ZnO	Salinity	10 mg/L	A rise in photosynthetic pigments, an increase in SOD and CAT activity, a decrease in proline, and a total increase in the amount of soluble sugars	[28]
*Musa acuminata*	SiO_2_	Salinity and water deficit	0, 200, 400, and 600 mg/L in vitro	Improved photosynthesis, preserved K+ and Na+ balance, decreased cell wall damage, and increased shoot growth and chlorophyll content	[29]
*Zea mays* L.	Polysuc-cinimide nanoparticles (PSI-NPs)	Cu	200 mg/L^−1^ through seeds	Increased the effectiveness of the water supply, which in turn raised the germination rate and percentage. Longer shoots, roots, and seedling fresh biomass all indicated PSI-NPs’ ability to reduce Cu stress	[30]
*Gossypium hirsutum* L.	Poly (acrylic acid)-coated cerium oxide nanoparticles (PNC)	Salinity	500 mg/L^−1^ through seedlings	In response to salinity stress, it was found that the length, vitality, fresh and dry weight, and root anatomical structure of seedlings all increased	[31]

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
