# Peer review of "Nanobionics: A Sustainable Agricultural Approach towards Understanding Plant Response to Heavy Metals, Drought, and Salt Stress"

_nanomaterials, 2023, doi:10.3390/nano13060974_

Round 1
Reviewer 1 Report
The present review deals with plant nanaobionics, a very important part of nanotechnology, applied to the use of nanoparticles to counteract various environmental stresses affecting, for example, plant germination and photosynthesis and crop productivity in general. Nanobionics activates plant defense mechanisms against stresses such as heavy metals, drought, salinity. Nanobionics is a new branch of nanobiotechnology and the present paper is therefore considered important for both the professional and lay public. It explains the main principles of plant nanobionics and presents some of the existing findings of a number of authors.
The paper also shows that there are many unresolved, mainly practical issues and that further studies are needed.
The article is suitable for publication in your journal, however in my opinion needs to be supplemented.
1. Nanobionics significantly offsets the negative effect of heavy metals on plants, especially by binds the toxic metal to the NP and reduces the metal's availability, according to the authors. However, the authors should, for example, clarify in the present paper whether the delivered nanoparticles, e.g. near the roots of plants, do not also have a negative effect, e.g. by adsorbing the delivered nutrients and preventing their entry into the plants, or whether they negatively affect the role of the delivered pesticides and, in particular, how they can affect soil microorganisms. These are basic questions that should be clearly explained.
2. The issue of nanobionics has been extensively studied recently and similar reviews already exist. The review presented here should be supplemented with commonly available more recent references. Quite randomly selected as possible examples, I give the following
Khatri and Rathore: Plant Nanobionics and Its Applications for Developing Plants with Improved Photosynthetic Capacity. DOI: 10.5772/intechopen.76815, 2018
Butnaria and Butu:Plant Nanobionics: Application of Nanobiosensors in Plant Biology. Plant Nanobionics. 2019 : 337–376.
Hassanzadeh et al: Nanobionics: From plant empowering to the infectious disease treatment. Journal of Controlled Release 349 (2022) 890-901
Ranjan et al: Nanobionics in Crop Production: An Emerging Approach to Modulate Plant Functionalities. Plants 2022, 11(5), 692
Author Response
Author responses to reviewer (1) comments
(Nanomaterials-2205087 R1)
To Reviewer #1
Remarks: The present review deals with plant nanaobionics, a very important part of nanotechnology, applied to the use of nanoparticles to counteract various environmental stresses affecting, for example, plant germination and photosynthesis and crop productivity in general. Nanobionics activates plant defense mechanisms against stresses such as heavy metals, drought, salinity. Nanobionics is a new branch of nanobiotechnology and the present paper is therefore considered important for both the professional and lay public. It explains the main principles of plant nanobionics and presents some of the existing findings of a number of authors.
The paper also shows that there are many unresolved, mainly practical issues and that further studies are needed.
Response: The authors are very thankful to the anonymous Reviewer for the appreciation, valuable suggestions, comments and scientific criticism of manuscript for its further improvement.
Remarks: Nanobionics significantly offsets the negative effect of heavy metals on plants, especially by binds the toxic metal to the NP and reduces the metal's availability, according to the authors. However, the authors should, for example, clarify in the present paper whether the delivered nanoparticles, e.g. near the roots of plants, do not also have a negative effect, e.g. by adsorbing the delivered nutrients and preventing their entry into the plants, or whether they negatively affect the role of the delivered pesticides and, in particular, how they can affect soil microorganisms. These are basic questions that should be clearly explained.
Response: Corrected and added text as per your suggestions.
Remarks: The issue of nanobionics has been extensively studied recently and similar reviews already exist. The review presented here should be supplemented with commonly available more recent references. Quite randomly selected as possible examples, I give the following
- Khatri and Rathore: Plant Nanobionics and Its Applications for Developing Plants with Improved Photosynthetic Capacity. DOI: 10.5772/intechopen.76815, 2018
- Butnaria and Butu:Plant Nanobionics: Application of Nanobiosensors in Plant Biology. Plant Nanobionics.2019 : 337–376.
- Hassanzadeh et al: Nanobionics: From plant empowering to the infectious disease treatment. Journal of Controlled Release 349 (2022) 890-901
- Ranjan et al: Nanobionics in Crop Production: An Emerging Approach to Modulate Plant Functionalities. Plants2022, 11(5), 692
Response: Added all the referred references.
All the suggestions and comments of the reviewer have been accepted by the authors and the manuscript has been corrected accordingly. A thorough internal reviews was also performed in the whole MS for possible improvement, changes highlighted in Track Change Format supplied MS. We hope the response meets the reviewer approval.

Reviewer 2 Report
Abstract: - line 30: „Thus, this review emphasizes the application of nanobionics on physiological traits…”- I would insert that „emphasizes the impact/influence of the application…”
Introduction:
- line 88: „Festuca ovina seeds are nonprime…”- What does is mean „nonprime”?- It should be „nanoprimed”, I think. Please check it.
- line 92: „increases”- it should be „increased”.
- line 96-97: „lining Triticum aestivum seeds with biosynthesized AgNPs (1 mg/L-1) increased the percentage of germination…”- this sentence is not clear for me. Please check it grammatically.
- line 98-100: I suggest to uset the verbs in past tense since these data have been already published.
- line 103-105: This sentence is confused grammatically. Please correct it.
- line 111: In this sentence the verb is missing, e.g. the recent review deals with the recent…”
- line 112: „the toxicity cause by HMs”- it must be written as „caused by”
- line 127: „through the cell membrane and cell wall”- I woul change the order, namely „through the cell wall and cell membrane”- in aspect of the NM.
- line 138-139: „The distribution of NPs on plant leaves can produce physical barriers depending on the species of plant”- What do You mean by this? What kind of barriers do You think of? Are these on the surface of the leaves? Do You think of the distribution of NPs on plant leaves within the mesophyll or the distribution on the epidermis?
- line 146-150: What is the reference You cited here?
- line 156: „has the possible to improve” – „possibility” would be better
- line 161-164: I miss the citations relating to these findings. Please check it.
- line 177: „ex vivo and in vivo”- These latin expressions must be written in italic.
- line 178: „Recent studies demonstrate…” I think it would be better to write: „Recent studies have demonstrated…”
- line 180: „Additionally, P3HT beads, particularly demonstrate that P3HT affects…”- I think that this part is not complete. I do not understand this.
- line 178-186: The findings You write about are already published, therefore it is better to use past tense not present. Please correct it grammatically.
- line 187: „this is made possible”- I suppose this sentence is not completed.
- line 195: „that up until it”- it is not clear for me.
- line 218: „mesophyll” is the right form.
- line 228: „toxic metal binds”- I think this is not true for all metals. The example must be more concrete.
- line 235: „it slows down…”- Does it refer to anatase?
- line 242: „accumulation” is better than „accumulating”.
- line 244: What does MSN mean?
- line 251-253: There is no citation.
- line 262-263: This sentence is not comprehensible.
- line 269-271: There is a repetition of “it is important to consider”. Please correct it.
- line 280: “reduce the length of the root apoplastic barrier”- What do You mean by length of the barrier?
- line 280-281: „Early apoplastic barrier development is induced by La2O3 NPs”- How did it happen concretely? E. g. did the endodermis (Casparian strip) develop closer to the root apex due to the NPs?
- line 293-295: „Although there have been studies that have demonstrated the presence of NPs along the Casparian strip…”- Which reports do support this statement? Please add some references.
References:
- Pleace check the titles of the articles cited grammatically.
- The latin names of the plants should be written in italic within the titles. Please check it.
Author Response
Author responses to reviewer (2) comments
(Nanomaterials-2205087 R1)
To Reviewer #2
The authors are very thankful to the anonymous Reviewer for the appreciation, valuable suggestions, comments and scientific criticism of manuscript for its further improvement.
Remarks: Abstract: - line 30: „Thus, this review emphasizes the application of nanobionics on physiological traits…”- I would insert that „emphasizes the impact/influence of the application…”
Response: Corrected.
Remarks: Introduction: line 88: „Festuca ovina seeds are nonprime…”- What does is mean „nonprime”?- It should be „nanoprimed”, I think. Please check it.
Response: Corrected.
Remarks: line 92: „increases”- it should be „increased”.
Response: Corrected.
Remarks: line 96-97: „lining Triticum aestivum seeds with biosynthesized AgNPs (1 mg/L-1) increased the percentage of germination…”- this sentence is not clear for me. Please check it grammatically.
Response: Corrected.
Remarks: line 98-100: I suggest to uset the verbs in past tense since these data have been already published.
Response: Corrected.
Remarks: line 103-105: This sentence is confused grammatically. Please correct it.
Response: Corrected.
Remarks: line 111: In this sentence the verb is missing, e.g. the recent review deals with the recent…”
Response: Added as per your suggestion
Remarks: line 112: „the toxicity cause by HMs”- it must be written as „caused by”
Response: Corrected.
Remarks: line 127: „through the cell membrane and cell wall”- I woul change the order, namely „through the cell wall and cell membrane”- in aspect of the NM.
Response: Corrected.
Remarks: line 138-139: „The distribution of NPs on plant leaves can produce physical barriers depending on the species of plant”- What do You mean by this? What kind of barriers do You think of? Are these on the surface of the leaves? Do You think of the distribution of NPs on plant leaves within the mesophyll or the distribution on the epidermis?
Response: It reduced the trapping of light by degrading the chlorophyll molecules present on the leaves.
Remarks: line 146-150: What is the reference You cited here?
Response: Corrected.
Remarks: line 156: „has the possible to improve” – „possibility” would be better
Response: Corrected.
Remarks: line 161-164: I miss the citations relating to these findings. Please check it.
Response: Added.
Remarks: line 177: „ex vivo and in vivo”- These latin expressions must be written in italic.
Response: Corrected as per your suggestions.
Remarks: line 178: „Recent studies demonstrate…” I think it would be better to write: „Recent studies have demonstrated…”
Response: Corrected as per your suggestions.
Remarks: line 180: „Additionally, P3HT beads, particularly demonstrate that P3HT affects…”- I think that this part is not complete. I do not understand this.
Response: Corrected as per your suggestions.
Remarks: line 178-186: The findings You write about are already published, therefore it is better to use past tense not present. Please correct it grammatically.
Response: Corrected as per your suggestions.
Remarks: line 187: „this is made possible”- I suppose this sentence is not completed.
Response: Corrected.
Remarks: line 195: „that up until it”- it is not clear for me.
Response: Corrected.
Remarks: line 218: „mesophyll” is the right form.
Response: Corrected.
Remarks: line 228: „toxic metal binds”- I think this is not true for all metals. The example must be more concrete.
Response: Corrected.
Remarks: line 235: „it slows down…”- Does it refer to anatase?
Response: Corrected.
Remarks: line 242: „accumulation” is better than „accumulating”.
Response: Corrected.
Remarks: line 244: What does MSN mean?
Response: Corrected.
Remarks: line 251-253: There is no citation.
Response: Deleted the text.
Remarks: line 262-263: This sentence is not comprehensible.
Response: Corrected as per your suggestions.
Remarks: line 269-271: There is a repetition of “it is important to consider”. Please correct it.
Response: Corrected as per your suggestions.
Remarks: line 280: “reduce the length of the root apoplastic barrier”- What do You mean by length of the barrier?
Response: It means the length of the outer cell layers of roots that reduces the oxic compounds from entering the root.
Remarks: line 280-281: „Early apoplastic barrier development is induced by La2O3 NPs”- How did it happen concretely? E. g. did the endodermis (Casparian strip) develop closer to the root apex due to the NPs?
Response: According to Yue et al. (2019) ZmPAL, ZmCCR2 and ZmCAD6, the core genes specific for biosynthesis of lignin, were up-regulated by 3-13 fold in roots exposed to 50 mg L-1 La2O3 NPs. However, ZmF5H was suppressed, indicating that lignin with S units could be excluded for the formed lignin in apoplastic barriers upon La2O3 NPs exposure.
Remarks: line 293-295: „Although there have been studies that have demonstrated the presence of NPs along the Casparian strip…”- Which reports do support this statement? Please add some references.
Response: Deleted the text.
Remarks: References: Pleace check the titles of the articles cited grammatically.
Response: Checked.
Remarks: The latin names of the plants should be written in italic within the titles. Please check it.
Response: Checked and corrected.
All the suggestions and comments of the reviewer have been accepted by the authors and the manuscript has been corrected accordingly. A thorough internal reviews was also performed in the whole MS for possible improvement, changes highlighted in Track Change Format supplied MS. We hope the response meets the reviewer approval.
